# ICAM: Interpretable Classification via Disentangled Representations and Feature Attribution Mapping

**Cher Bass**
Kings College London
London, UK, WC2R 2LS
cher.bass@kcl.ac.uk

**Mariana da Silva**
Kings College London
London, UK, WC2R 2LS

**Carole Sudre**
Kings College London
London, UK, WC2R 2LS

**Petru-Daniel Tudosiu**
Kings College London
London, UK, WC2R 2LS

**Stephen M. Smith**
University of Oxford
Oxford, UK, OX1 2JD

**Emma C. Robinson**
Kings College London
London, UK, WC2R 2LS
emma.robinson@kcl.ac.uk

## Abstract

Feature attribution (FA), or the assignment of class-relevance to different locations in an image, is important for many classification problems but is particularly crucial within the neuroscience domain, where accurate mechanistic models of behaviours, or disease, require knowledge of all features discriminative of a trait. At the same time, predicting class relevance from brain images is challenging as phenotypes are typically heterogeneous, and changes occur against a background of significant natural variation. Here, we present a novel framework for creating class specific FA maps through image-to-image translation. We propose the use of a VAE-GAN to explicitly disentangle class relevance from background features for improved interpretability properties, which results in meaningful FA maps. We validate our method on 2D and 3D brain image datasets of dementia (ADNI dataset), ageing (UK Biobank), and (simulated) lesion detection. We show that FA maps generated by our method outperform baseline FA methods when validated against ground truth. More significantly, our approach is the first to use latent space sampling to support exploration of phenotype variation.

## 1 Introduction

Brain images present a significant resource in the development of mechanistic models of behaviour and neurological/psychiatric disease as they reflect measurable neuroanatomical traits that are heritable, present in unaffected siblings and detectable prior to disease onset [10]. Nevertheless, for complex disorders, features of disease remain subtle, variable [21, 38] and occur against a back-drop of significant natural variation in shape and appearance [16, 27].

Traditional approaches for brain image analysis compare data in a global average template space, estimated via smooth (and ideally diffeomorphic) deformations [2, 11, 12, 14, 16, 37]. This, however, typically ignores cortical heterogeneity and may smooth out sources of variation [11, 16] in ways which limit interpretation [7]. Tools are still required to distinguish between features of population variability and specific discriminative phenotypic features.

Deep learning is state-of-the-art for many image processing tasks [17] and has shown strong promise for brain imaging applications such as healthy tissue and lesion segmentation [8, 13, 25, 36]. However, there is growing need for greater accountability of networks, especially within the medical domain. Several approaches for feature attribution (FA) [3, 30, 33, 40, 43] have been proposed which return

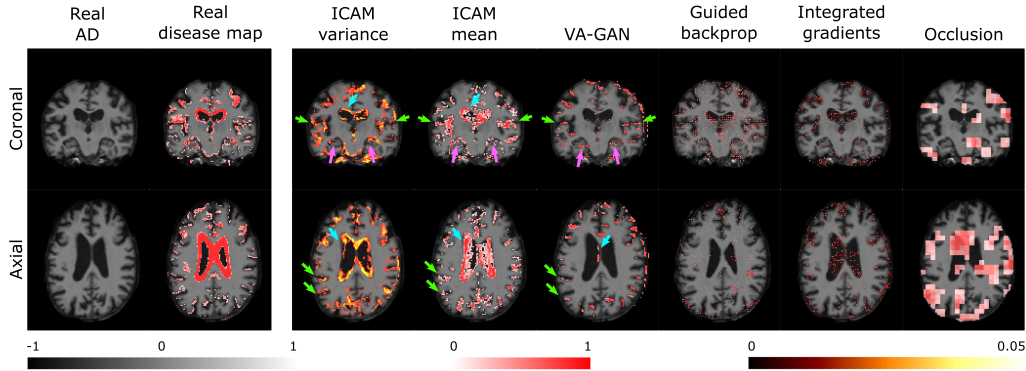

Figure 1: ADNI comparisons of feature attribution (FA) maps. ICAM is the first known method able to generate variance and mean FA maps in test time, and shows good detection of the ventricles (blue arrows), cortex (green arrows), and hippocampus (pink arrows) when compared with the ground truth disease map. Baseline methods perform sub-optimally in comparison, with VA-GAN generating the second best FA maps.

the most important or salient features for a prediction, after training a network for classification. However, applying a method post-hoc instead of explicitly training an interpretable model has shown to be insufficient at detecting all discriminative features of a class, especially in medical imaging [5]. Recently, approaches have been proposed which use generative models to translate images from one class to another [22, 48], and in Baumgartner et al. [5] this was adapted to create a difference map between Alzheimer's (AD) and Mild Cognitive Impairment (MCI) subjects. While it was able to detect more salient features in comparison to previous methods, it was still unable to identify much of the variability between the AD and MCI subjects. Other restrictions of this approach are that it assumes knowledge of label classes at test time, and that it does not have a latent space that can be sampled. This limits the interpretation and generates a deterministic output at test time.

In this paper we aim to improve on the current feature attribution methods by developing a more interpretable model, and thus more meaningful feature attribution maps. We propose ICAM (Interpretable Classification via disentangled representations and feature Attribution Mapping), a framework which builds on approaches for image-to-image translation [28] to learn feature attribution by disentangling class-relevant *attributes* (attr) from class-irrelevant *content*. Sharp reconstructions are learnt through use of a Variational Autoencoder (VAE) with a discriminator loss on the decoder (Generative Adversarial Network, GAN). This not only allows classification and generation of an attribution map from the latent space, but also a more interpretable latent space that can visualise differences between and within classes. By sampling the latent space at test time to generate a FA map, we demonstrate its ability to detect meaningful brain variation in 3D brain Magnetic Resonance Imaging (MRI).

In particular the specific contributions of the method are as follows:

1  We describe the first framework to implement a translation VAE-GAN network for simultaneous classification and feature attribution, through use of a shared attribute latent space with a classification layer, which supports rejection sampling and improved class disentanglement, relative to previous methods [28].

2  This supports exploration of phenotypic variation in brain structure through latent space visualisation of the space of class-related variability, including the study of the mean and variance of feature attribution map generation (see example in Fig. 1).

3  We demonstrate the power and versatility of ICAM using extensive qualitative and quantitative validation on 3 datasets including; Human Connectome Project (HCP) with lesion simulations, Alzheimer's Disease Neuroimaging Initiative (ADNI), and UK Biobank datasets. In addition, our code, which has been released on GitHub at `https://github.com/CherBass/ICAM`, extends to multi-class classification and regression tasks.

Table 1: Comparison of baseline methods.

| Method | post-hoc | classification | generative model | cyclic | variance analysis |
|---|---|---|---|---|---|
| Grad-CAM [40] | ✓ | ✓ | | N/A | |
| Guided backprop [42] | ✓ | ✓ | | N/A | |
| Integrated gradients [43] | ✓ | ✓ | | N/A | |
| Occlusion [47] | ✓ | ✓ | | N/A | |
| DRIT [28] | | | ✓ | ✓ | |
| VA-GAN [5] | | | ✓ | | |
| ICAM (our method) | | ✓ | ✓ | ✓ | ✓ |

## 2 Related works

### 2.1 Feature attribution methods

The most commonly used approach for feature attribution follows the training of a classification network with importance or saliency mapping (Table 1, rows 1-4). These typically analyse the gradients or activations of the network, with respect to a given input image, and include approaches such as Gradient-weighted Class Activation Mapping (Grad-CAM) [40], SHAP [30], DeepTaylor [33], integrated gradients [43], guided backpropagation (backprop) [42], and Layer-wise backpropagation (LRP) [3]. In contrast, perturbation methods such as occlusion [47] change or remove parts of the input image to generate heatmaps, by evaluating its effect on the classification prediction. Most of these methods, however, provide coarse and low resolution attribution maps. More importantly, they depend on a network trained prior to applying a particular FA method, and will perform sub-optimally if the network learnt to focus on only the most salient features relevant to a class (e.g. focus on a dog's face for natural image prediction, but not on its tail or other distinctive features); this is common with classification networks.

### 2.2 Generative models

An alternative approach is the use of generative models, specifically GANs and VAEs, where a common application, image-to-image translation, has been used successfully in many different domains [22, 48, 20, 29, 24, 28], including medical imaging [5, 4, 6, 9]. Of these, Lee et al. [28], in particular, developed a domain translation network called DRIT (Fig. 2b), which constraints features specific to a class, through encoding separate class-relevant (attribute) and class-irrelevant (content) latent spaces, and employing a discriminator (Table 1, row 5).

Separately, Baumgartner et al. [5], developed a conditional GAN-based approach, called VA-GAN, for feature attribution, using domain translation between 3D MRI of brains with AD and MCI (Table 1, row 6). In this work, a mapping $M$ is learnt, which translates an AD input image towards the MCI class (Fig. 2a), resulting in sharp reconstructions and realistic difference maps that overlap with ground truth outcomes, where available. One constraint of VA-GAN, however, is that it requires image class labels to be known *a priori*. And, in the absence of a latent space, it can only produce

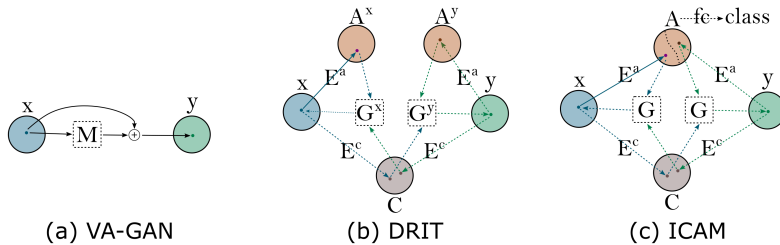

Figure 2: Comparison of domain mapping methods. (a) VA-GAN translates images of domain x to y. (b) DRIT can translate between domains x and y through a shared content space $C$, and separate attribute spaces $A^x$ and $A^y$. (c) ICAM uses shared content $C$ and attribute $A$ spaces to translate between domains, which allows a classification layer to be applied to the attribute space $A$.

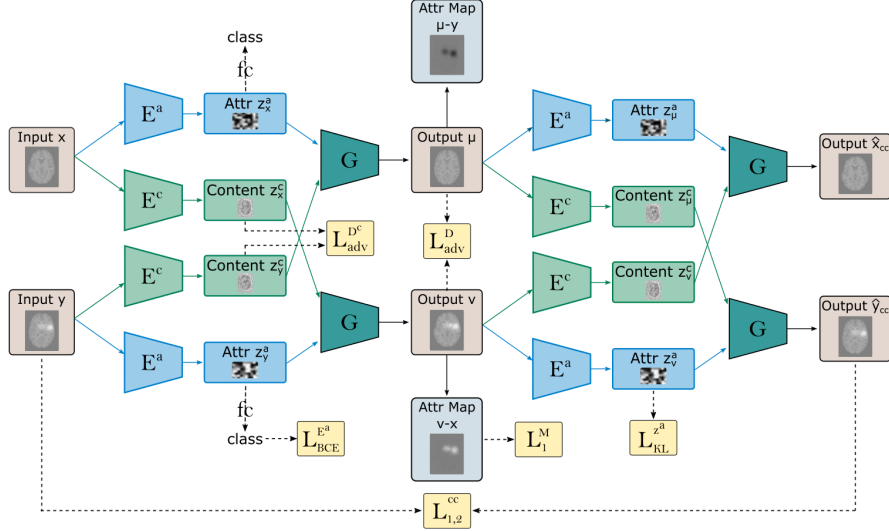

Figure 3: Overview of method. An example of how ICAM performs classification with attribute map generation for 2 given input images x and y (of class 0 [brain slice without lesions] and 1 [brain slice with simulated lesions], respectively). Note that $L_{adv}^D$ is applied to both real and generated images, and that not all losses are plotted (see Equation 5 for full objective).

a single deterministic output for an input image, which limits the interpretation in comparison to methods with a latent space.

In this paper, we therefore extend the intuitions of these models to create one framework which allows simultaneous classification and feature attribution, using a more interpretable model. In particular, we use a VAE-GAN to encode a class-relevant attribute latent space which is shared between classes, and thus allows classification and the visualisation of differences between and within classes (Fig. 2c). Table 1 summarises the advantages of ICAM relative to other feature importance, saliency, and generative visualisation methods.

## 3 Methods

### 3.1 Method overview

The goal of our framework is to learn a classifier that encodes inputs of different classes into a separable latent space, and a generator that synthesizes FA maps with all class-relevant salient features. We use a VAE-GAN with an adversarial content latent space, which learns about class-irrelevant information, and an attribute latent space, which learns about class-relevant information. The overview of the framework is shown in (Fig. 3). The approach is comprised of the following components:

A **content encoder** $\{E^c\}$ encodes class-irrelevant information using a shared content latent space $\{z_x^c, z_y^c \in C\}$, via the application of a **content discriminator** $\{D^c\}$, whose goal is to discriminate between the classes or domains. We refer to domain or class interchangeably, in which the same meaning is implied. For input images $\{x, y\}$ of classes $\{c_x\}$ and $\{c_y\}$, respectively, the goal of the content encoder $\{E^c\}$ is to fool the discriminator to classify an image incorrectly, and to make the content latent space appear the same, regardless of the class $(E^c : x \to z_x^c), (E^c : y \to z_y^c)$. An **attribute encoder** $\{E^a\}$ learns all relevant class information, and classifies between domains $(E^a : x \to z_x^a \to c_x), (E^a : y \to z_y^a \to c_y)$ using a fully connected/ dense layer that is applied to the shared attribute (class) latent space $\{z_x^a, z_y^a \in A\}$. The **generator** $\{G\}$ learns to synthesise an image conditioned on both the content and attribute latent spaces $(G : \{z_x^c, z_x^a\} \to \hat{x}), (G : \{z_y^c, z_y^a\} \to \hat{y})$, as well as to translate between these domains, by swapping the content latent space; this is possible since the content latent space is class invariant $(G : \{z_y^c, z_x^a\} \to \mu), (G : \{z_x^c, z_y^a\} \to v)$.

Translating the domains enables the visualisation of differences between the two classes, using a **feature attribution map** ($\{M_x = v - x\}, \{M_y = \mu - y\}$). Finally, the **domain discriminator** $\{D\}$ learns to distinguish between generated and real images, and to classify the two domains, which gives a clearer training signal for the generator.

Our network architectures uses 2D or 3D convolutional layers (kernel size = 3 or 4), ResNet layers including basic, down, and deconvolutional blocks with instance normalisation. A key feature of the network architecture is encoding the latent space as a 2D or 3D vector, instead of a 1D vector as is commonly seen in VAEs. For example, for a latent space size of 80, a 1D vector = [80], 2D vector = [8,10]. This allows the network to encode spatial, and shape related information in the latent space, which is important in brain imaging. A detailed diagram of our encoder-generator architecture (for a 3D input) is shown in supplementary section A.1.

## 3.2 Content and attribute spaces

Our approach disentangles the two image domains into a shared content space $\{C\}$, and attribute space $\{A\}$. The content latent space aims to encode class irrelevant information: features common to both domains (e.g. location of structures, and folds of the brain). The attribute latent space aims to map the remaining domain-specific information onto shared latent space $\{A\}$, thus enabling classification using all salient features, and cross-domain translation.

**Content loss.** To achieve domain disentanglement, we employ a common content encoder for both domains ($\{E^c : x \rightarrow C\}, \{E^c : y \rightarrow C\}$). The content latent space $\{C\}$ is fed into a content discriminator, $\{D^c\}$, which outputs the image class probability. The content discriminator $\{D^c\}$ aids the representation to be disentangled, by aiming to distinguish between domains (classes) of the encoded latent spaces $\{z_x^c\}$ and $\{z_y^c\}$. Inversely, the content encoder $\{E^c\}$ aims to encode a representation whose domain cannot be distinguished by the content discriminator, and thus forces the representation to be mapped to the same space $\{C\}$, similarly to Lee et al. [28].

This class adversarial content loss can be expressed as:

$$
\begin{aligned}
L_{adv}^{D^c} = \ &\mathbb{E}_{z_x^c}[\log D^c(E^c(x)) + \log(1 - D^c(E^c(x)))] \\
+ \ &\mathbb{E}_{z_y^c}[\log D^c(E^c(y)) + \log(1 - D^c(E^c(y)))].
\end{aligned}
\tag{1}
$$

An L2 regularisation was added to prevent explosion of gradients, while Gaussian noise was added to the last layer of the content encoder to prevent vanishing of the content latent space.

**Classification loss.** Classification is performed through extending the attribute latent space using a fully connected layer with binary cross entropy loss $L_{BCE}^{E^a}$, to encourage the separation of the domains within the shared attribute latent space $\{A\}$.

**VAE loss.** We employ a latent variable model, where we place a Gaussian prior over the latent variables and train using variational inference, by applying the Kullback Leibler (KL) loss $L_{KL}^{z^a}$.

**Latent regression loss.** We impose an additional loss on the attribute latent space in order to encourage invertible mapping between the image and the latent space, which also aids the cyclic reconstruction. We sample an attribute latent vector $z_r^a$ from a Gaussian distribution, and attempt to reconstruct it:

$$
L_1^{z^a} = \|E^a(G(E^c(x), z_r^a)) - z_r^a\|_1.
\tag{2}
$$

**Rejection sampling of the attribute latent space.** Disentanglement is further encouraged through 'rejection' sampling of the attribute latent space during training by checking the class of a randomly sampled vector using the attribute encoder's classification layer. Samples are rejected if they belong to the wrong class, which stabilises optimisation of translation by passing the generator samples of the expected class. For example, when translating an image of class A to class B, sampling a vector of class A would be considered incorrect, whereas sampling a vector of class B would be considered correct. This also applies in a multi-class setting. This further allows translation (using a single image) in test time, by first encoding an input image (Fig. 4a), and then sampling from the space until the opposite class is sampled (Fig. 4b), by checking the random vector's class using the classifier.

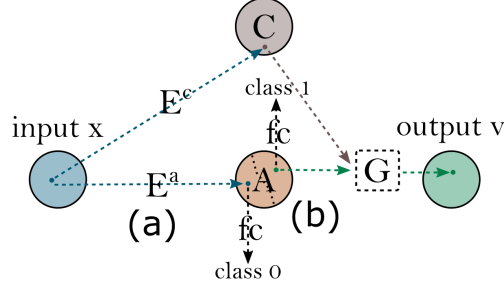

Figure 4: Rejection sampling during training/ testing. Using ICAM, translation can be achieved using a single input image, in addition to translating between 2 images. (a) An input image is encoded into content and attribute spaces, and is passed through the classifier to identify its class (0 in this example). (b) Attribute space A is then randomly sampled until a random vector of the opposite class is sampled (class 1 in this case), by checking its class using the classifier. The newly sampled vector is passed to the generator along with the encoded content space to achieve translation between class 0 and 1.

By sampling multiple times, we can also get mean and variance FA maps during test time (See an example in Fig. 1 and 5), a visualisation approach previously not possible in other feature attribution methods, as they do not have a latent space with a classification layer.

### 3.3 Generation and feature attribution

**Feature attribution (FA) map loss.** To visualise differences between the translated images $\{v, \mu\}$ and the original images $\{x, y\}$, we use a feature attribution map $\{M\}$. This aims to retain only class-related differences between two images (or two locations in the attribute latent space) by subtracting the content from the translated output ($\{M_x = v - x\}, \{M_y = \mu - y\}$). Generation is regularised through an L1 loss ($L_1^M = \|M(\ )\|_1$,) which encourages $\{M\}$ to reflect a small feasible map, which leads to a realistic translated image.

**Domain loss.** The domain loss combines a domain adversarial loss, $L_{adv}^D$, (to discriminate between real and generated images, and encourage realistic image generation) and a classification binary cross entropy loss, $L_{BCE}^D$ (to encourage generation of images of the expected class).

**Reconstruction loss.** To facilitate the generation, we apply an L1 and L2 loss to the reconstructed images $\{\hat{x}, \hat{y}\}$ ($L_1^{rec}$), and the cyclically reconstructed images $\{\hat{x_{cc}}, \hat{y_{cc}}\}$ ($L_1^{cc}$). The cycle consistency term also allows training with unpaired images.

$$L_{1,2}^{rec} = \mathbb{E}_{x,y}[\|G(E^c(x), E^a(x)) - x\|_{1,2} + \|G(E^c(y), E^a(y)) - y\|_{1,2}], \tag{3}$$

$$L_{1,2}^{cc} = \mathbb{E}_{x,y}[\|G(E^c(v), E^a(\mu)) - x\|_{1,2} + \|G(E^c(\mu), E^a(v)) - y\|_{1,2}]. \tag{4}$$

This means the **full objective function**[1] of our network is:

$$\min_{G,E^c,E^a} \max_{D,D^c} \lambda_{D^c} L_{adv}^{D^c} + \lambda_D L_{adv}^D + \lambda_{D_{BCE}} L_{BCE}^D + \lambda_{BCE} L_{BCE}^{E^a} + \lambda_{KL} L_{KL}^{z^a}$$
$$+ \lambda_M L_1^M + \lambda_{z^a} L_1^{z^a} + \lambda_{rec}(L_1^{rec} + L_1^{cc} + L_2^{rec} + L_2^{cc}). \tag{5}$$

## 4 Results

We evaluate the performance of ICAM through studies on three datasets to perform 1) ablation studies on 2D simulations; 2) evaluation of the accuracy of the generated attribution maps (using ground

truth disease conversion maps derived from the ADNI dataset); 3) exploration of the flexibility of the approach for investigating phenotypic variation (using healthy ageing data from UK Biobank).

## 4.1 Comparison methods and metrics

We compared our proposed approach against a range of baselines in our experiments. For a fair comparison, we use the same training, validation and testing datasets. In particular, we compared against **Grad-CAM**, **guided Grad-CAM** [40], **guided backprop** [42], **Integrated gradients** [43], **Occlusion** [47], **VA-GAN** [5], and **DRIT++** [28]. These methods were applied to both a simple 3D ResNet (Table 3), and to the attribute encoder ($E^a$) of ICAM (Table 4). We compare against 2 variations of the DRIT++ network, the original, $DRIT_{z_8}$, and with increased attribute latent space size ($DRIT_{z_{80}}$, size of 80 rather than 8), which is the same as ICAM. We also compared against different variations of our network, **ICAM**, including: $ICAM_{DRIT}$, uses ICAM architecture, but all the same losses as in DRIT; $ICAM_{BCE}$, which adds the classification BCE loss to the attribute latent space and rejection sampling during training; $ICAM_{FA}$, which adds the FA map loss; and finally $ICAM$, adding the l2 loss and thus containing all described losses in this paper. For further details on comparison methods, refer to supplementary section A.5.

Networks are compared using accuracy score for classification, and normalised cross correlation (NCC) between the absolute values of the attribution maps and the ground truth masks (e.g. the lesion masks in the HCP lesion simulations, or the disease effect maps in ADNI). The positive NCC (+) compares the lesion mask to the attribution map when translating between class 0 (e.g. no lesions, or MCI) to 1 (e.g. lesions, or AD), and vice versa for the negative NCC (-). Values reported are the mean and standard deviation across the test subjects.

## 4.2 HCP 2D ablation experiments

Many neurological and psychiatric disorders show variability in presentation and symptoms, meaning that it is common to observe differences in the imaging phenotype. In our ablation experiments, we used the HCP dataset [16, 45], with T2 MRI data, for simulating lesions in MRI brain slices, to mimic this type of variability. We use the original data as class 0 (no lesions), and then create cortical 'lesions' which appear in the 'disease' class 1, with different frequencies. During training, we take 2D axial slices from the centre of the brain (to which all lesions are constrained by design), so that we can compare against DRIT, which is a 2D network. For further details on the HCP dataset, see supplementary section A.2.

Ablation results are displayed in Table 2, where we show that $ICAM_{DRIT}$ network (i.e. ICAM architecture, with the same losses as DRIT) performs similarly to DRIT, whilst having a much more compact architecture. Due to memory constraints, only ICAM supports extension to 3D (with 1.8M rather than 2.6M trainable parameters, making the 3D encoder-decoder network of DRIT $1.4\times$ bigger). Furthermore, we show that the addition of different components of the network improves performance further (see NCC scores, Table 2, rows 3-6), with the best overall performance achieved by the full ICAM network (Table 2, row 6). In addition, we observe that qualitatively ICAM performs better when interpolating between and within classes (Fig. B.2 in the supplementary). ICAM FA maps appear to be smoothly changing when interpolating between the lesion and no lesion class, while in DRIT the FA maps are similar across the interpolation, suggesting that some of the class information is encoded in the content latent space. Further, we observe that ICAM is able to both add and remove lesions simultaneously while interpolating within the lesion class, whereas DRIT is only able to remove lesions. Overall, this indicates that ICAM has achieved better separation in

Table 2: Ablation experiments using the 2D HCP dataset with lesion simulations.

| Network | Accuracy | NCC (-) | NCC (+) |
|---|---|---|---|
| $DRIT_{z_8}$ | N/A | $0.346 \pm 0.080$ | $0.243 \pm 0.050$ |
| $DRIT_{z_{80}}$ | N/A | $0.380 \pm 0.081$ | $0.272 \pm 0.068$ |
| $ICAM_{DRIT}$ | N/A | $0.385 \pm 0.094$ | $0.265 \pm 0.062$ |
| $ICAM_{BCE}$ | 0.899 | $0.316 \pm 0.102$ | $0.325 \pm 0.100$ |
| $ICAM_{FA}$ | 0.900 | $0.353 \pm 0.104$ | $\mathbf{0.333 \pm 0.082}$ |
| $ICAM$ | **0.950** | $\mathbf{0.435 \pm 0.092}$ | $0.332 \pm 0.098$ |

Table 3: ADNI experiments.

| Network | NCC (-) | NCC (+) |
|---|---|---|
| Guided Grad-CAM [40] | $0.244 \pm 0.047$ | $0.339 \pm 0.068$ |
| Grad-CAM [40] | $0.321 \pm 0.059$ | $0.461 \pm 0.086$ |
| Occlusion [47] | $0.360 \pm 0.037$ | $0.354 \pm 0.057$ |
| Integrated gradients [43] | $0.378 \pm 0.064$ | $0.404 \pm 0.059$ |
| Guided backprop [42] | $0.541 \pm 0.054$ | $0.532 \pm 0.052$ |
| VA-GAN [5] | $0.653 \pm 0.142$ | N/A |
| ICAM | $\mathbf{0.683 \pm 0.097}$ | $\mathbf{0.652 \pm 0.083}$ |

the attribute latent space, and is further supported by the tSNE plots within class 1 (Fig. B.1 in the supplementary).

## 4.3 ADNI experiments: Ground-truth evaluation of feature attribution maps

We use the longitudinal ADNI dataset [23], with T1 MRI data, as in Baumgartner et al. [5], to evaluate feature attribution maps generated by ICAM and other baseline methods. DRIT [28] was not used as the network design cannot scale to 3D due to memory constraints. ADNI contains paired examples for which images that are acquired before and after conversion to an AD state; where the intermediate state between healthy cognition and AD is known as MCI. We used these subjects in our validation and test sets, to calculate the ground truth disease map (i.e. AD-specific brain atrophy patterns) for each subject, which we then compared against the FA maps of each method to compute the NCC score. See supplementary materials section A.3 for further details on the dataset.

We found that ICAM outperforms VA-GAN when comparing the NCC metric (Table 3), and that they both perform better than occlusion, integrated gradients, Grad-CAM, guided Grad-CAM and guided backprop. The FA maps generated by the VA-GAN in our comparisons differ to the original results of VA-GAN [5], and these differences might be accounted by a more strict data selection process (we picked images acquired with 3T only, and did not combine with 1.5T acquired images as in Baumgartner et al. [5]), which resulted in a smaller training size (5778 vs 931 subjects for training in VA-GAN and this work, respectively), and small differences in the pre-processing. In our experiments, we observe that ICAM is able to detect most of the real disease effects in ventricles, cortex, and hippocampus, but that VA-GAN only detects some of these differences (Fig. 1, and Fig. B.3 in the supplementary). Both ICAM and VA-GAN generate higher resolution, and more interpretable FA maps in comparison to occlusion, integrated gradients, Grad-CAM, guided Grad-CAM and guided backprop, suggesting that using generative models, instead of a simple classification CNN, might be a better approach for detecting more discriminative features, and phenotypic variability.

Finally, we tested whether ICAM FA generation performs better than FA methods (occlusion, integrated gradients and guided backprop) applied to ICAM's attribute network $E^a$. We found that as expected ICAM's FA generation achieves better performance (using the NCC metric) than the FA methods (Table 4).

## 4.4 Biobank experiments

We used T1 MRI data from the UK Biobank [1, 32], a collection of brain imaging data of mostly healthy subjects between the ages of 44-80 years old, to study phenotypic variation that occurs during ageing. To use this dataset for classification, we split the data into 2 classes, of young (class 0,

Table 4: ADNI experiments with ICAM $E^a$.

| Network | NCC (-) | NCC (+) |
|---|---|---|
| ICAM-$E^a$ Occlusion | $0.235 \pm 0.067$ | $0.310 \pm 0.047$ |
| ICAM-$E^a$ Integrated gradients | $0.269 \pm 0.054$ | $0.289 \pm 0.046$ |
| ICAM-$E^a$ Guided backprop | $0.295 \pm 0.056$ | $0.301 \pm 0.044$ |
| ICAM | $\mathbf{0.683 \pm 0.097}$ | $\mathbf{0.652 \pm 0.083}$ |

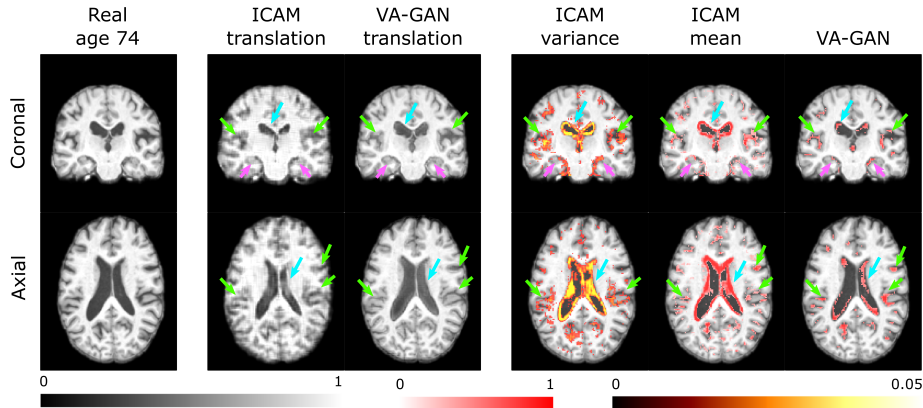

Figure 5: Biobank comparisons: modelling healthy ageing (translation of an old subject to young). We show that ICAM has visibly better detection in the hippocampus (pink arrows), ventricles (blue arrows), and cortex (green arrows). Further, ICAM is able to change the shape of different brain regions, whereas VA-GAN is only able to make minor adjustments in pixel intensities.

45-60 years) and old subjects (class 1, 70-80). For further details on the dataset see section A.4 of supplementary materials.

We achieved 0.943 accuracy on the test dataset with ICAM on age classification, but cannot compare to VA-GAN, as it cannot perform class prediction. We compare the feature attribution maps and translated images against VA-GAN (Fig. 5, and Fig. B.5 in the supplementary). While we do not have ground truth maps for a comparison in Biobank, it is commonly described within the neuroscience domain that hippocampal atrophy [34], decrease in cortical thickness, and the enlargement of the ventricles are often observed with ageing, and are indicative of neurodegeneration [18, 35]. We looked for this phenotypic variation in our qualitative analysis. For ICAM we plot variance, and mean FA maps (computed through multiple sampling of the attribute latent space in test time); for VA-GAN we show only FA as it can only generate a single deterministic output. We found that VA-GAN was able to detect some variation in the ventricles and cortex, but that ICAM was able to detect more variability in the ventricles (blue arrows), cortex (green arrows), and in the hippocampus (purple arrow). Interestingly, the variance map detects higher variability in the ventricles and the cortex, which may reflect heterogeneity in the ageing process. We also note that while ICAM is able to change the shape of different brain regions in translation (e.g. shrinking the ventricles, or enlarging the cortical folds), VA-GAN is only able to change the pixel intensities, and does not appear to change the shape. This is a significant advantage for ICAM compared to previous methods, as shape is an important phenotype in many other medical datasets [15, 26], and thus is likely to generalise well.

To demonstrate the added value of the components introduced in ICAM (in comparison to DRIT), we also compared against a basic version of our network, $ICAM_{DRIT}$, and show that (1) the full version of ICAM, but not $ICAM_{DRIT}$, is able to separate the old and young subjects in the latent space, and is thus (2) able to interpolate between subjects in the latent space (see Figs. B.4 and B.6 in the supplementary).

## 5   Conclusion

In this work we developed a novel framework for classification with feature attribution. We demonstrate that our method achieves better performance on the HCP with lesion simulation, ADNI and UK Biobank datasets, compared to previous work. To our knowledge, it is the only approach able to generate FA maps directly from the attribute (class-relevant) and content (class-irrelevant) latent spaces, and its highly interpretable latent space allows detailed analysis of phenotypic variability, by studying the variance and mean FA maps. Finally, our code supports generalisation between 2D and 3D image spaces, and extensions to multi-class classification and regression.

## Broader Impact

There is growing evidence that deep learning tools have the potential to improve the speed of review of medical images and that their sensitivity to complex high-dimensional textures can (in some cases) improve their efficacy relative to radiographers [19]. A recent study by Google DeepMind [31] suggested that deep learning systems could perform the role of a second-reader of breast cancer screenings to improve the precision of diagnosis relative to a single-expert (which is standard clinical practice within the US).

For brain disorders the opportunities and challenges for AI are more significant since the features of the disease are commonly subtle, presentations highly variable (creating greater challenges for physicians), and the datasets are much smaller in size in comparison to natural image tasks. The additional pitfalls that are common in deep learning algorithms [46], including the so called 'black box' problem where it is unknown why a certain prediction is made, lead to further uncertainly and mistrust for clinicians when making decisions based on the results of these models.

We developed a novel framework to address this problem by deriving a disease map, directly from a class prediction space, which highlights all class relevant features in an image. Our objective is to demonstrate on a theoretical level, that the development of more medically interpretable models is feasible, rather than developing a diagnostic tool to be used in the clinic. However, in principle, these types of maps may be used by physicians as an additional source of data in addition to mental exams, physiological tests and their own judgement to support diagnosis of complex conditions such as Alzheimer's, autism, and schizophrenia. This may have significant societal impact as early diagnosis can improve the effectiveness of interventional treatment. Further, our model, ICAM, presents a specific advantage as it provides a 'probability' of belonging to a class along with a visualisation, supporting better understanding of the phenotypic variation of these diseases, which may improve mechanistic or prognostic modelling of these diseases.

There remain ethical challenges as errors in prediction could influence clinicians towards wrong diagnoses and incorrect treatment which could have very serious consequences. Further studies have shown clear racial differences in brain structure [41, 44] which if not sampled correctly could lead to bias in the model and greater uncertainty for ethnic minorities [39]. These challenges would need to be addressed before any consideration of clinical translation. Clearly, the uncertainties in the model should be transparently conveyed to any end user, and in this respect the advantages of ICAM relative to its predecessors are plain to see.

## Acknowledgments and Disclosure of Funding

The work of E.C.R. and C.B. was supported by the Academy of Medical Sciences/the British Heart Foundation/the Government Department of Business, Energy and Industrial Strategy/the Wellcome Trust Springboard Award [SBF003/1116] and E.C.R., C.B. and S.M.S. are supported by Wellcome Collaborative Award [215573/Z/19/Z]. PD.T. was supported by the EPSRC Research Council, part of the EPSRC DTP [EP/R513064/1]. M. DS. would like to acknowledge funding from the EPSRC Centre for Doctoral Training in Smart Medical Imaging [EP/S022104/1].

The ADNI data used in this work was funded by the Alzheimer's Disease Neuroimaging Initiative (ADNI) (National Institutes of Health Grant U01 AG024904) and DOD ADNI (Department of Defense award number W81XWH-12-2-0012). The UK Biobank data was accessed under Application Number 8107. The HCP data was provided by the Human Connectome Project, WU-Minn Consortium (Principal Investigators: David Van Essen and Kamil Ugurbil; 1U54MH091657) funded by the 16 NIH Institutes and Centers that support the NIH Blueprint for Neuroscience Research; and by the McDonnell Center for Systems Neuroscience at Washington University.

We would also like to thank Kai Arulkumaran and Ankur Handa for valuable comments and suggestions during manuscript writing.

The Authors declare that there is no conflict of interest.

## Footnotes

[1] see supplementary materials section A.6 for $\lambda$ values during training.

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
