[Supplementary Material]

# Supplementary Materials of ICAM: Interpretable Classification via Disentangled Representations and Feature Attribution Mapping

**Cher Bass**
Kings College London
London, UK, WC2R 2LS
cher.bass@kcl.ac.uk

**Mariana da Silva**
Kings College London
London, UK, WC2R 2LS

**Carole Sudre**
Kings College London
London, UK, WC2R 2LS

**Petru-Daniel Tudosiu**
Kings College London
London, UK, WC2R 2LS

**Stephen M. Smith**
University of Oxford
Oxford, UK, OX1 2JD

**Emma C. Robinson**
Kings College London
London, UK, WC2R 2LS
emma.robinson@kcl.ac.uk

## A  Methods

### A.1  Network Architecture

Our encoder-decoder architecture for a 3D input is shown in Fig. A.1. The architecture for a 2D input is the same, only using 2D convolutions and a 2D attribute space. Here, an input image is encoded using 2 shared networks, the attribute encoder $E_a$, and the content encoder $E_c$, and then is reconstructed or translated (to another class) using the generator, $G$.

The key components of the attribute encoder include using down ResNet blocks (with average pooling, and leaky ReLU activation) for encoding the input image into a relatively large 3D latent space of size $8 \times 10 \times 8$ (in the 3D case), as opposed to a 1D vector, which is commonly seen in Variational Autoencoders (VAEs). We also added a fully connected layer to the attribute latent space to enable classification. In early development, we found that using a 1D vector in the latent space was insufficient for encoding the required class information for brain imaging, and observed that some class information was instead encoded in the content encoder, which is meant to be class invariant. Using a sufficiently large 2D or 3D vector (depending on the input) helped with addressing this problem.

The goal of the content encoder is to encode a class-irrelevant space, which allows translation between classes. The key components of the content encoder is using 2 down convolutional blocks (with instance normalisation, and ReLU activation), followed by 4 basic ResNet blocks (with instance normalisation, and ReLU activation), and finally a Gaussian noise layer. The basic ResNet blocks aids the encoding of a class-irrelevant space, and the Gaussian layer prevents the space from becoming zero.

Our generator takes in as input the content and attribute latent spaces. The attribute is first upsampled ($\times 4$, with nearest neighbors) to the same size as the content latent space, concatenated, and then combined using several basic ResNet blocks. Finally, we use deconvolutional blocks (transpose convolution with kernel size of 4, followed by average pooling, layer normalisation [2], and a ReLU activation) to upsample to the original input size.

In addition, not shown in Fig. A.1, our domain discriminator contains 6 convolutional layers with leaky ReLUs (kernel size = 3, stride = 2), followed by 2 additional convolutional layers (kernel size = 1, stride = 1), and adaptive average pooling for each class output, and real/ fake output. Our content

Figure A.1: Network Architecture for 3D inputs.

discriminator contains 3 convolutional layers with leaky ReLUs (kernel size = 3, stride = 2), followed by an additional convolutional layer (kernel size = 4, stride = 1), adaptive average pooling, and a final fully connected layer for class output.

## A.2 The Human Connectome Project (HCP) dataset

Imaging phenotype variability is common in many neurological and psychiatric disorders, and is an important feature for diagnosis. This type of variation was simulated in Baumgartner et al. [3] for a simple 2D case where class 0 was simulated with no features and class 1 was simulated as two sub-types, with one feature in common and then a second feature, which appears in a different location in each sub-population.

We use the HCP dataset with T2 MRI volumes to take this simulation further, and to create cortical features or 'lesions' which appear in the 'disease' class with different frequencies (Fig. A.2). Specifically, eight regions were selected from the HCP parcellation [5]. These were defined as binary masks for cortical surface meshes and then mapped back to the cortical volume (with a 2mm thickness), using HCP 'workbench' tools [10], followed by intensity scaling to match intensity of the cortical spinal fluid, and finally by a Gaussian blur filter.

Every simulated example includes common regions with a further locations were selected as having lesion or no lesion using a random number generator. The regions selected are; $MT_R$, $MT_L$ (medial temporal area), $OP1_R$, $OP1_L$ (posterior opercular cortex), $v23ab_R$, $v23ab_L$ (posterior cingulate cortex), $9a_R$, $9a_L$ (medial prefrontal cortex), where R and L are right and left, respectively. $MT_R$ and $MT_L$ were selected as common regions, and appear in every subject. To be able to compare against DRIT, 2D networks were trained on 2D axial slices from the centre of the brain (to which all regions are constrained by design). Since there is a lot of variability between subjects, not all the lesions appear in the selected slice for each subject, and so there is further variability in lesion appearance.

Prior to training, the T2 images were bias corrected, brain extracted and linearly aligned (for full details on image acquisition and pre-processing see [4]). Images were normalised in range [0, 1] per subject, and resized to $128 \times 160 \times 128$ voxels, and sliced into 2D axial images, of size $128 \times 160$. Data was randomly split for training, validation and testing using an 80/10/10 ratio (712, 88, 88 subjects each, respectively), consistently for all networks.

Figure A.2: Example of a 2D MRI axial slice from the HCP dataset with and without lesions. We note that the simulated lesions are of similar pixel intensities to the CSF. This is often observed in pathological lesions, and can make them challenging to detect.

## A.3 Alzheimer's Disease Neuroimaging Initiative (ADNI) dataset

Used also in Baumgartner et al. [3] the longitudinal ADNI dataset [6] supports ground-truth evaluation of the feature attribution (FA) maps as it contains a subset of paired examples for which images are acquired before and after conversion to full a AD state; where the intermediate state between healthy cognition and full dementia is known as mild cognitive impairment (MCI). The age average and standard deviation for AD subjects is $74.95 \pm 8.1$, for MCI subjects is $72.26 \pm 7.9$, and for test subjects is $73.47 \pm 7.2$. The 3T acquired T1 MRI volumes were N4 bias corrected [19] using simpleitk, brain extracted using freesurfer [14] and rigidly registered to the MNI space using Niftyreg [12]. Images were normalised in range $[-1, 1]$ per subject, and resized to $128 \times 160 \times 128$ voxels. We split the dataset into AD and MCI classes, with 257 and 674 volumes used in training, respectively. For testing, 61 subjects which convert from MCI to AD (i.e. paired subjects) are used. A further 61 conversion subjects are used for validation. To compute the disease maps, these paired subjects were in addition rigidly aligned to each other, and the difference between them was the disease map for that pair. Finally, all disease and FA maps were masked to ensure that the returned NCC values reference brain tissue only.

## A.4 UK Biobank dataset

UK Biobank data included [1, 11] 11,735 T1 3D MRI volumes, selected from two age bins 45-60 years (class 0, on average $54.6 \pm 3.4$ years) and 70-80 years (class 1, on average $73.0 \pm 2.2$ years). T1 image processing (see also [1]) involved bias correction using FAST [21], brain extraction using BET [16] and linear registration to MNI space, using the FLIRT toolbox [7]. Our young subjects are separated into training, validation, and testing sets with 6706, 373 and 372 in each, respectively. Our older subjects are separated into training, validation, and testing sets with 3856, 214 and 214 in each, respectively. The input into the networks is resized to $128 \times 160 \times 128$ voxels, and normalised in range $[0, 1]$, per subject.

## A.5 Comparison methods

We compare our proposed approach against a range of baselines in our experiments. For a fair comparison, we train and test all methods on the same training, validation and testing datasets.

**Grad-CAM, guided Grad-CAM [15], guided backpropagation (backprop) [17], integrated gradients [18] and occlusion [20].** We trained a simple 3D ResNet with 4 down ResNet blocks, and a fully connected layer for classification. We then used the captum library [8] implementation of

Grad-CAM, guided Grad-CAM, guided backprop, integrated gradients and occlusion to generate the feature attribution maps for each method.

Guided backprop [17] is a gradient-based method that computes the gradients with respect to an input image. More specifically it determines which pixels affect the prediction the most, by propagating only positive error signals (i.e. by applying ReLU to to the error during the backward pass).

Grad-CAM [15] is gradient-based saliency method that computes the gradients of the target output with respect to the final convolutional layer of a network. The layer activations are weighted by the average gradient for each output channel and the results are summed over all channels to produce a coarse heatmap of prediction importance for each class. Guided Grad-CAM is simply the combination of the results of Grad-CAM and guided backprop.

Integrated gradients [18] is another method of analysing the gradient of the prediction output with respect to features of the input. It is defined as the integral of the gradients along the straight line path from a given baseline to the input image. A series of images are interpolated between the baseline (e.g. matrix of 0s) and the original image, and the integrated gradients are given by the integration of the computed gradients for all the images in the series.

Occlusion [20] is a perturbation-based method that involves replacing portions of an image with a block of a given baseline value (e.g. 0), and computing the difference in output. A heatmap is formed using the difference between the output probability attributed to the original volume and the probability computed for the occluded volume, for different positions of the occlusion block across the input image.

Grad-CAM was implemented on the last convolutional block of the ResNet, with a size of $4 \times 5 \times 4$, and was up-sampled to the input size for visualization. For the implementation of integrated gradients we considered a baseline volume with constant value of 0, and the integral was computed using 200 steps. Occlusion was implemented using occlusion blocks with value 0, size $10 \times 10 \times 10$ and stride 5.

**FA methods on the attribute encoder of ICAM ($E^a$).** We further performed guided backprop [17], occlusion [20] and integrated gradients [18] on the attribute encoder of ICAM ($E^a$).

**DRIT [9].** In our ablation experiments, we use the 2D network DRIT++ described in Lee et al. [9]. Because we aim to use the network for feature attribution, instead of its original goal of domain translation, we had to make some changes to the original network. The aim of the ablation experiments were to assess the different components of ICAM, so we used the most comparable version of DRIT. In particular, we use the DRIT++ network, which has a shared generator, but use an unconditional version (i.e. without the input of class label) of the network, so that it is comparable to ICAM. We also generate the FA map in the same way.

**VA-GAN [3].** We used the VA-GAN network for feature attribution, as described in the original paper.

**Model selection.** For VA-GAN, $ICAM_{DRIT}$ and $ICAM$, the last model is selected in the Biobank experiments. In all other experiments, the models selected are based on the best model result on the validation dataset, using the NCC score. For Grad-CAM, guided Grad-CAM, guided backprop, integrated gradients and occlusion, as the FA maps are only generated after a network is trained, we could not select a model based on its performance with the NCC score, during training/ validation. We instead selected the best model based on the accuracy classification score on the validation dataset, to prevent the effect of overfitting.

### A.6 Training details

We used PyTorch [13] Python package in all of our deep learning experiments, and trained using NVIDIA TITAN GPUs. We trained our networks in a similar fashion to Lee et al. [9]. During training in each iteration, the content discriminator is updated twice, followed by the update of the encoders, generators, and domain discriminators (i.e. each training iteration uses 3 batches to perform these updates). For each update of the generator, an input is selected for each class (e.g. 2 inputs including class 0 and 1) to achieve translation. In addition, each input is encoded and translated to the opposite

class by randomly sampling the attribute latent space, and obtaining an appropriate class, using the classifier.

In all experiments, unless otherwise stated, we used the following hyperparameters during training of ICAM networks: learning rate for content discriminator = 0.00004, learning rate for the rest = 0.0001, Adam optimiser with betas = (0.5, 0.999), $\lambda_{D^c} = 1, \lambda_D = 1, \lambda_{BCE} = 10, \lambda_{KL} = 0.01, \lambda_M = 10, \lambda_{z^a} = 1, \lambda_{rec} = 100, \lambda_{D_{BCE}} = 1$ for discriminator optimisation, and $\lambda_{D_{BCE}} = 5$ for generator optimisation. We do not use augmentation techniques in any of our experiments.

In the UK Biobank experiments, we trained all networks for 50 epochs. In the HCP 2D ablation and ADNI experiments, all networks (including VA-GAN and DRIT) were trained for 300 epochs. In the ADNI experiments, because we had a limited dataset, we further refined ICAM with updated lambdas ($\lambda_{rec} = 10$, and $\lambda_{BCE} = 20$) for another 200 epochs. We could not refine VA-GAN any further because generator and discriminator losses went to zero during training, often after 150 epochs.

**Baseline methods.** For training VA-GAN, and DRIT, we used the default parameters as provided in the original papers and publicly released code repositories. For Grad-CAM, integrated gradients, and occlusion, the classifier network was trained with learning rate of 0.0001, SGD with momentum of 0.9, for 50 epochs, and using a weighted BCE loss to account for class-unbalanced training data. Since the model converged by 50 epochs, we did not train for any longer.

# B    Results

## B.1    HCP experiments

In our additional HCP experiments (see section A.2 for dataset details), we show that ICAM can interpolate between classes (Fig. B.2, left), showing a smooth interpolated result, as well as capturing the large majority of the lesions in both addition and removal, whereas DRIT does not visibly demonstrate smooth interpolation, and is only able to do removal of lesions effectively. This suggests that the content latent space has not been completely disentangled, and that it might encode some class information (about the lesions). That is most likely caused by the network architecture, as

Figure B.1: tSNE plots for ICAM and DRIT methods using the HCP test dataset.

**Interpolation between classes**   **Interpolation within class 1**

input a          input a                    input a          input a

ICAM            DRIT                        ICAM            DRIT

0                                1          -1           0            1

Figure B.2: Interpolation between classes 0 (no lesions) and 1 (lesions), and within class 1. Red, addition of lesions; blue, removal of lesions.

encoding the attribute latent space as a 1D vector, instead of a 2D vector as with ICAM, might not be sufficient to encode spatial information about the lesions. In addition, ICAM shows visibly better interpolation within class 1 in the latent space (Fig. B.2, right). ICAM is able to both add and remove lesions simultaneously during interpolation, while DRIT is only able to remove lesions.

Finally, ICAM demonstrates clear separation between class 0 (no lesions) and 1 (lesions) in its latent space (Fig. B.1, top, left). In addition, without explicitly giving information about the different lesions, ICAM is able to cluster several of the lesions in the latent space (shown via the circles), when encoding tSNE for class 1 (lesions with 8 different locations) examples (Fig. B.1, bottom, left). DRIT is also very separable between classes 0 and 1 (Fig. B.1, top, right), but does not demonstrate separation within class 1 (Fig. B.1, bottom, right).

## B.2 ADNI experiments

We show additional examples (see section A.3 for dataset details) of comparisons between ICAM and baseline methods in Fig. B.3. In general, we observe ICAM achieves visibly better detection compared to baseline methods, with the variance map of ICAM most sensitive to variability in the cortex (green arrows), ventricles (blue arrows), and hippocampus (pink arrows), and appeared to be the most similar to the real disease maps (e.g. rows 3-4).

We note that ICAM and VA-GAN seem to detect some differences which do not appear in the ground truth disease map (e.g. hippocampal atrophy, pink arrows, row 5). Some of this may relate to measurement noise, for example motion is known to be a significant challenge in dementia datasets, and ADNI scans are acquired with variable acquisition parameter. Overall ICAM variance maps flag up more evidence of disease than VA-GAN; however it is important to stress that MCI-AD conversion is not a binary process. These maps are therefore likely picking up heterogeneity in the relative timing and disease progression in these subjects.

## B.3 Biobank experiments

In our additional Biobank experiments (see section A.4 for dataset details), we performed translation with very old ($> 79$ years), and young ($< 50$ years) subjects (Fig. B.5). VA-GAN is not a cyclic network and therefore does not have an example for young to old translation. We found very high detection in relevant brain regions for both ICAM and VA-GAN for translation of old subjects to young. VA-GAN appeared to detect more hippocampal differences than ICAM when comparing to ICAM's FA mean maps, but appeared similar when comparing to the variance maps (rows 1 and 3). In addition, ICAM was able to detect much higher variability in the ventricles and was able to change the shape of the brain, while VA-GAN was only able to make minor adjustments in pixel intensities. While we found weaker detection in our young to old translation (Fig. B.5, bottom), many of the expected regions were still highlighted in the FA maps, and we even observed changes in brain shape, including enlargement of the ventricles (blue arrows) and decrease in cortical thickness (green arrows).

To demonstrate the effectiveness of the added components in $ICAM$, we compared against a baseline version of our network, $ICAM_{DRIT}$, as DRIT cannot scale to 3D. We performed interpolation between old and young subjects (Fig. B.6), and found that with $ICAM_{DRIT}$, the FA maps are similar across the interpolation. It is possible that the attribute and content spaces have not been disentangled, and that some class relevant information was encoded in the content space. In contrast, $ICAM$ demonstrates smooth interpolation between the classes, and thus is likely to have disentangled age in the attribute latent space. Additional evidence for this is the tSNE plots of $ICAM$, and $ICAM_{DRIT}$ (Fig. B.4), where we see separation between old and young subjects for $ICAM$, but no separation for $ICAM_{DRIT}$.

Finally, we report image generation quality for Biobank; although we stress that the objectives of this model was generation of disease maps (for which we show clear improvements) rather than image generation. Nevertheless, the Fréchet Inception Distance (FID) score (which measures the similarity between two datasets) indicates that VA-GAN outperforms ICAM (with respective scores 14.01 and 38.05; lower is better). A better result for VA-GAN is to be expected as VA-GAN is a U-Net style network, with high level skip connections, whereas ICAM receives much more downsampled features that support the learning of a meaningful latent space for improved disease map generation.

Figure B.3: Additional AD to MCI ADNI comparisons for FA map generation in 3 subjects. Blue arrows, ventricles; green arrows, cortex; pink arrows, hippocampus.

Figure B.4: tSNE plots for Biobank on age classification.

Figure B.5: Biobank translation results for translation of old to young (top) and translation of young to old (bottom) in 4 subjects. Blue arrows, ventricles; green arrows, cortex; pink arrows, hippocampus.

Figure B.6: Biobank interpolation between class 0 (young) and 1 (old) for $ICAM$ and $ICAM_{DRIT}$.

## B.4   Yosemite weather experiments

To demonstrate that the method can generalise across domains, and specifically to natural image datasets, we performed further experiments on the Yosemite dataset [22], a natural image dataset with summer and winter weather scenes. We aimed to show that while ICAM was optimised for 3D medical image datasets, it can still be applied to natural image datasets and produce realistic results. We found that ICAM translated the images reasonably well (Fig. B.7), by for example adding snow in the summer to winter translation, or by adding more green to the trees in the winter to summer translation. However, we note that ICAM does not generate images that are as crisp as in DRIT++ [9]. ICAM produced images are blurrier and contained checkerboard artifacts on certain objects. Since we used the default parameters for training on Yosemite, the quality of generations could be improved with hyperparameter turning.

Figure B.7: Yosemite dataset translation results with ICAM.

## B.5 ICAM reproducibility experiments

The reproducibility of ICAM's FA maps must be validated for ICAM to have clinical potential. Indeed, to demonstrate that ICAM produces consistent results across the same or similar inputs, we ran two experiments. 1) We applied ICAM on (unseen) images acquired at multiple time points for one subject with AD diagnosis, and compared the outputs. Fig. B.8a shows that ICAM generates very similar FA maps for all images (despite them being independently acquired and processed) suggesting the method is reproducible, consistent, and that anatomy is preserved. Further evidence is provided by Fig. B.8b, which shows that repeat runs of ICAM on Biobank data generate very similar FA mean and variance maps despite taking different samples from the latent space, producing low variance ($\leq 0.0003$) across $\times 10$ experiments.

Figure B.8: ICAM reproducibility experiments. (a) ICAM was applied to images of the same ADNI subject at multiple time-points. We show that the mean FA maps are similar across all inputs. (b) ICAM was applied to the same Biobank subject 10 times. We display one example of the mean and variance FA map (column 2), and the variance across the 10 experiments (column 3).