[Reviews · NeurIPS 2020]

Review 1

Summary and Contributions: The paper presents a cycle-gan inspired approach which simultaneously classifies an image, maps it into a class-specific and class-agnostic content representation and maps it onto an explanation domain.

Strengths: The paper presents an interesting approach of latent space mapping with enforced distantanglement. (POST-rebuttal): Some questions are left open indeed, however I believe that the open questions are not of such a severity, that it would disqualify the approach as a whole and justify to reject the paper. The open questions seem to be rather on the level: there are matters to be clarified why or how the method works. Given that it has an evaluation and shows improved results over several baselines and it is a complex method on 8 pages, in my view it is worth for a wider audience to read this paper, and if desired, do their own research on the open questions.

Weaknesses: Reproducibility: Are there enough details to reproduce the major results of this work?: R replied no because important for a network of such a complexity: initialization details of the training is missing (any data augmentation?). they must be included in the supplement. that is more important than a broader impact discussion. Comparison of explanations is made against some weak baselines: IG is known to be very noisy, Grad-CAM is known to be very lowly resolved. What is missing is: Guided-gradCAM, Guided BP (these two are very easy to implement ... , either via an autograd function or backward hook on relus) and possibly LRP. All those three have in test-time speed in the order of a second. The first two are known to be as fast as a gradient backpropagation step. Also missing is the NCC evaluation for those methods, in particular as they were shown to perform well on brain imaging tasks (see citations below). (POST-rebuttal): they have made measurements on GB and Guided GB, however unexpectedly on E^c, while the class relevant content is in E^a, and the measurement of explanations would need to be done on E^a and on the classifier heads.

Correctness: looks correct.

Clarity: yes

Relation to Prior Work: Some contributions need to be discussed: https://link.springer.com/chapter/10.1007%2F978-3-030-33850-3_1 https://www.frontiersin.org/articles/10.3389/fnins.2019.01321/full?report=reader

Reproducibility: No

Additional Feedback: Furthermore this is an implicit consequence of a model that predicts and then explains a superset of the prediction. (POST-rebuttal): Actually it explains the encoder features, where the classification decision is only a projected subset of the encoder features f via y=w \cdot f + b, and all encoder content orthogonal to w gets ignored in the classification, but appears still in the explanation in their model. it should be investigated (1) how good is the faithfulness of the explanation to the classification itself, means the prediction as opposed to groud truth (2) whether the better correlation to human / simulator ground-truth comes from the fact that they explain E^a -output rather only the classification head, which is a one-dim projection of the E^a feature map


Review 2

Summary and Contributions: The paper proposes ideas from disentanglement to derive relevant from irrelevant features of a discriminative model.

Strengths: * The application domain is important, and it is essential to point out that GRad-CAM type methods are not informative not sufficient nor proper for some application domain such as medical imaging.

Weaknesses: * Although the method targets medical imaging. The authors should have tested the general framework in situated or bench mark dataset where the "real signal" is well controlled and the report the how well the method can detect the true signal.

Correctness: * I am not convince by the motivation of the paper. The authors argue that training an interpretable model result in better performance. Such argument contradicts with the current understanding in the field that enforcing "interoperability" compromise the prediction and this is why there more interest in "explaining" the Blackbox rather than enforcing interoperability of the model. * I am also not convinced by the construction of the model. Given two instances, the idea is to swap the class-irrelevant feature and reconstruct the image again and compare it with the original; the difference would be the authors argue, the relevant information. Consider brain disease where normal anatomical changes are irrelevant to the class while atrophy in certain areas do (this is the case about Alzheimer's disease). If one swaps the irrelevant class information, the difference images mostly characterize the real anatomical differences cluttering the true signal (atrophy).

Clarity: * The paper is quite confusing. In the method section there are several mentioning of the "domain" as if the paper was written for domain adaptation. It is not clear what they mean by "domain". * The notation of the paper is not consistent and not well explained. For example are $E^c$ and $E_c$. * The symbols in the figures are not explained (for example fig 2). Also the quality of the figures could have been much better. * The quantitative results are very limited. The authors need to test this framework on a simulated or benchmark data where the ground truth is known or easy to spot. * The method uses VAE-GAN, one the metric to report is how realistic the resulting images are. Nothing is reported in that regard.

Relation to Prior Work: * Please search the literatture for alternative expandability methods for medical images. I agree with the authors that the grad-cam is not proper for medical images.

Reproducibility: No

Additional Feedback: * I recommend the authors to rewrite the paper in a way that all notation are clearly explain. * Please remove all "domain" terms. * Improve the quality of images. * Report some results how well VAE-GAN work in term of producing realistic looking results.


Review 3

Summary and Contributions: This paper proposes a training structure for learning feature attribution maps during training time by disentangling the class relevant information within the input and generating images with modified class. The method is validated on medical datasets.

Strengths: The work shows strong empirical results and a novel architecture: 1. Empirically, the work shows strong improvement over previous benchmarks considered on the medical datasets tested. 2. Empirically, this method may be regularizing and increasing robustness of the model compared to a simple classification loss, although this remains under-explored. 3. Architecturally, the proposed method of generating class-changed encodings with constant 'content' information while incorporating a classification sub-network appears to be novel, and it is somewhat surprising that this conjunction of so many different losses was able to train to convergence. 4. Methodologically, this approach of doing pixel-wise comparisons of class-translated images is a novel approach to feature attribution (If the DRIT method also compared to feature attribution methods then this is not novel).

Weaknesses: Since this direction of research is quite novel and complex, it is of paramount importance to validate the methodology carefully. The greatest weakness of the work is inadequate evaluation of the proposed methodology. Open questions include: 1. (The rebuttal has not responded to this concern) Are explanations faithful to model behavior? As this method involves counterfactual augmentation, it remains unclear whether the generated FA maps are faithful to the marginal contribution of input features to prediction. For example, the generation of the FA maps relies on sampling in the latent space, if the sampling is not dense enough it is possible that the 'nearest semantically meaningful neighbor' (this remains to be defined) is not sampled. 1b. (The rebuttal partially responded to this concern) To what extent are the observed increases in NCC caused by changes in model behavior as opposed to improved interpretability method fidelity? To clarify this question one could apply grad-CAM/Integ. Grads to the classification sub-network of the proposed model. POST-REBUTTAL: The rebuttal ran grad-cam etc. on the content encoder network, but it remains unclear to me why they chose to test interpretability on the content encoder as opposed to the classifier sub-model? Ideally all of the reasonable uses of grad-cam etc. would be reported in the appendix. 2. (The rebuttal responded to this concern adequately) Are counterfactual augmentations realistic? This could be addressed by asking medically trained experts to evaluate generated images. 3. (The rebuttal responded to this concern partially) To what extent are these results generalizable? Can this method generate realistic counterfactual augmentations for imagenet? for CIFAR? The rebuttal provided a sample, but since there is no accompanying NCC metric to show, it is important to have a random sample. It seems to me that the couple images shown by the rebuttal do not provide strong evidence that this method will work in those contexts. This is a severe limitation given the venue is NeurIPS and not a biomedical venue. Addressing these shortcomings is critical to understanding whether (and how) the proposed method works as claimed. On the other hand, as the paper stands the validity of the methodology remains unproven.

Correctness: Almost all of the paper appeared to be error free. There may be a problem with the domain loss. The text claims the domain loss enforces distinguishing between real and fake images. In Figure 3 two fake images are compared by the domain loss. It's unclear to me whether the figure or text is incorrect or perhaps just misleadingly phrased?

Clarity: The paper does a commendable job of making clear a very involved model structure and loss. At some points I did have to reference the supplement to understand the paper: 154-163: Rejection sample refers to sampling a 'wrong' class. Wrong class is not defined in a multi-class classification setting this could be clarified. 171-173: As mentioned above, it remains unclear to me what the L^D_adv is doing and how it is defined? 224-244: It should be clarified in the main text that Grad-CAM etc. are applied to a _different_ model, NOT the classification sub-model of the proposed model.

Relation to Prior Work: Yes, the comparison to VA-GAN and DRIT are clear and Figure 2 is useful for this purpose as well. The relationship with other interpretability methods could be clarified. The paper claims the method can be used as a feature-attribution method, but it is very different from gradient-based (and other post-hoc) interpretability methods. See Weaknesses discussion above.

Reproducibility: Yes

Additional Feedback: If this work generalizes to other datasets it may be a very valuable contribution. Most urgently points 1 and 1b from the Weaknesses section above must be addressed. Without this the method can be meaningfully termed a 'generative' method, but cannot be called an 'interpretability' or 'feature attribution' method.


Review 4

Summary and Contributions: The authors proposed ICAM, which extends DRIT++ and make it suitable for feature attribution. They applied the method to synthetic datasets and brain images, showing that the method can capture the difference associated with the development of AD and ageing.

Strengths: The authors explained the rationale behind different components of the method, and empirically showed the effect of them through ablation experiments. The feature attribution through latent space as a single model is significant and novel contribution.

Weaknesses: (Not addressed in author feedback) In terms of utility and relevance to other tasks, many datasets for medical image classification tasks have class imbalance. Due to the training process that uses paired examples from each class, the method may be impacted more heavily than gradient based method from class imbalance.

Correctness: (This point was partially addressed) For empirical validation, the author used the difference between before and after developing AD as ground truth. However, without comparison to control patients who did not develop AD for the same time difference, it is still possible that the feature attribution and classification may be coming from ageing. Average age for different classes should be included and discussed. (Post rebuttal) The author reported the average age for each group, but without providing any details on the data to properly address the concern. Especially, it is not clear how images from the same patients are included in the experiments.

Clarity: The paper is clearly written.

Relation to Prior Work: The paper includes direct comparison to previous methods, e.g. GradCAM, VA-GAN and DRIT.

Reproducibility: Yes

Additional Feedback: (This point was not addressed in author feedback) One minor point is that I think the explanation for L^D_{adv} could be improved. It is not just discriminating between real and fake images in general, but it is to discriminate them in each class.

[Author Response · NeurIPS 2020]

We thank the reviewers for their insightful comments and agree with all points related to clarity of the terminology, notation, and image quality. We would correct all these points for any camera ready copy of the manuscript.

R2 is correct in suggesting that the ultimate goal of machine learning for healthcare should be explainable models. However, interpretability and explainability need not be mutually exclusive. For complex neurological and psychiatric disorders, where the causes are unclear and symptoms are highly heterogeneous, development of explainable models may be non-trivial, arguably requiring prior hypotheses of the types of variation expected. It is here where interpretable models such as ICAM can help. At the same time, ICAM may be positioned alongside other radiological support tools, such as Mckinney Nature 2016, which are already showing potential for clinic use. Indeed, the results from this paper, have led to new clinical collaborations for the development of tools for pre-surgical planning of epilepsy.

Nevertheless, as R3 suggests (points 1&2), the anatomical validity of the counter-factual augmentation must be validated for ICAM to have clinical potential. Accordingly we ran two experiments 1) we applied ICAM on Alzheimer's (unseen) images acquired at multiple time points for the same subject and compared the outputs. Fig. 1a shows that ICAM generates very similar FA maps for all images (despite them being independently acquired and processed) suggesting the method is reproducible, consistent and that anatomy is preserved. Further evidence is provided by Fig. 1b, which shows that repeat runs of ICAM on Biobank data generate very similar maps despite taking different samples from the latent space, producing low variance ($\leq 0.0003$) across $\times 10$ experiments. However, we also note that we would seek clinical verification of these findings for any camera ready copy. And, in response to R2 related points, we swap the attribute (class) latent space that is 3D by design, to allow class relevant spatial information to be encoded (e.g. brain atrophy), whereas subject-specific brain anatomy is encoded by the 3D content (class-irrelevant) latent space.

We find R2's request for reporting image generation quality reasonable; although we stress that the objectives of this model was generation of disease maps (for which we show clear improvements) rather than image generation. Nevertheless, the Fréchet Inception Distance (FID) score (which measures the similarity between two datasets) indicates that VA-GAN outperforms ICAM (with respective scores $14.01$ and $38.05$; lower is better). A better result for VA-GAN is to be expected as VA-GAN is a U-Net style network, with high level skip connections, whereas ICAM receives much more downsampled features that support the learning of a meaningful latent space for improved disease map generation.

We agree with R1 that more thorough details of the training process should go in the supplement. We also plan to upload a cleaned and commented version of the code to Github with examples (all data sources are open source and available). We also appreciate R1 literature suggestions and request for more benchmarking. Accordingly we performed guided backpropagation (GB) and guided Grad-CAM (G-CAM) on the ADNI data set and found that, while GB offers improved performance (over other baselines) of $0.541 \pm 0.05$ Normalised Cross-Correlation (NCC) (-) and $0.532 \pm 0.05$ NCC(+), it is still worse than VA-GAN and ICAM (see Table 3 in paper). G-CAM does not perform as well, with scores of $0.244 \pm 0.05$ NCC(-) and $0.339 \pm 0.07$ NCC(+). In addition, in response to R3 point 1b we applied feature attribution methods (including GB, integrated gradients (IG) and occlusion (OC)) to the encoder network of ICAM $E^c$. We report GB of $0.296 \pm 0.06$ NCC(-) and $0.301 \pm 0.04$ NCC(+), IG of $0.269 \pm 0.05$ NCC(-) and $0.289 \pm 0.05$ NCC(+), and OC of $0.235 \pm 0.07$ NCC(-) and $0.310 \pm 0.05$ NCC(+). These scores are not as good as the feature attribution (FA) maps generated by ICAM's generator network. As discussed in the paper, these baselines methods suffer from being low-resolution and, by design, ignore phenotypically variable features. By contrast ICAM is designed to generate high resolution disease maps sensitive to all areas of pathology.

Finally, R4 correctly highlights that the age distribution of the MCI and AD training groups should be reported. The age average and standard deviation is $74.95 \pm 8.1$, and $72.26 \pm 7.9$, for AD and MCI subjects, respectively. In response to R2's point on testing on a simulated dataset with ground-truth, we did do this for a simulated lesion HCP dataset (see Table 2 in the paper, and Figs. A.3 & B.2 in the supplement). However, we agree with R2 &R3 (point 3) that benchmarking on a natural image dataset (where the ground truth is easier to visually verify) would be informative. Therefore in Fig. 1c we show that ICAM works well (but could be further improved with hyperparameter optimisation) on the Yosemite dataset (as used in the DRIT paper), and will be added to the supplement.

Figure 1: Additional experiments.

[Meta-Review · NeurIPS 2020]

This paper proposes a model for simultaneous classification and feature attribution in the context of medical image classification. The model uses GAN to learn two representations from pairs (x, y) of input images of different classes. One representation is class-relevant (z^a, a for attribution) and the other is class-irrelevant (z^c, c for content). The class-relevant representation is used for classification. Both representations are fed to a generator G to synthesize images so as to achieve domain translation. While G(z^c_x, z^a_x) approximately recovers x, G(z^c_x, z^a_y) is a translation of x into the class of y (what x would look like if it were from the class of y). Consequently, the difference G(z^c_x, z^a_y) - x can be used as a visualization (attribution) of the difference of the two classes in the domain of x. Empirically, the work shows strong improvement over previous benchmarks considered on the medical datasets tested. There was a long series of extensive discussions (12 long posts). One reviewer remains unconvinced about the novelty and some technical issues, while two other reviewers are strongly supportive. Overall, the idea seems interesting and the work is solid. The fact that it has generated so much debates among the reviewers is a good sign. It shows that the model is worth a further attention and study.